

# Predicting the risk of acute respiratory failure among asthma patients—the A2-BEST2 risk score: a retrospective study

Yanhong Qi[1,*], Jing Zhang[2,*], Jiaying Lin[1,3], Jingwen Yang[1,3], Jiangan Guan[2], Keying Li[1], Jie Weng[1,4], Zhiyi Wang[1,4], Chan Chen[2,4] and Hui Xu[1,4]

[1] General Practice, The Second Affiliated Hospital and Yuying Children's Hospital of Wenzhou Medical University, Wenzhou, China

[2] Geriatric Medicine, The First Affiliated Hospital of Wenzhou Medical University, Wenzhou, China

[3] General Practice, Taizhou Women and Children's Hospital of Wenzhou Medical University, Taizhou, China

[4] Wenzhou Medicial University, Sourthern Zhejiang Institute of Radiation Medicine and Nuclear Technology, Wenzhou, China

[*] These authors contributed equally to this work.

Corresponding author
Hui Xu, xuhui2580@126.com

## ABSTRACT

**Objectives**. Acute respiratory failure (ARF) is a common complication of bronchial asthma (BA). ARF onset increases the risk of patient death. This study aims to develop a predictive model for ARF in BA patients during hospitalization.

**Methods**. This was a retrospective cohort study carried out at two large tertiary hospitals. Three models were developed using three different ways: (1) the statistics-driven model, (2) the clinical knowledge-driven model, and (3) the decision tree model. The simplest and most efficient model was obtained by comparing their predictive power, stability, and practicability.

**Results**. This study included 398 patients, with 298 constituting the modeling group and 100 constituting the validation group. Models A, B, and C yielded seven, seven, and eleven predictors, respectively. Finally, we chose the clinical knowledge-driven model, whose C-statistics and Brier scores were 0.862 (0.820–0.904) and 0.1320, respectively. The Hosmer-Lemeshow test revealed that this model had good calibration. The clinical knowledge-driven model demonstrated satisfactory C-statistics during external and internal validation, with values of 0.890 (0.815–0.965) and 0.854 (0.820–0.900), respectively. A risk score for ARF incidence was created: The $A_2$-$BEST_2$ Risk Score ($A_2$ (area of pulmonary infection, albumin), BMI, Economic condition, Smoking, and $T_2$ (hormone initiation Time and long-term regular medication Treatment)). ARF incidence increased gradually from 1.37% (The $A_2$-$BEST_2$ Risk Score $\leq 4$) to 90.32% ($A_2$-$BEST_2$ Risk Score $\geq 11.5$).

**Conclusion**. We constructed a predictive model of seven predictors to predict ARF in BA patients. This predictor's model is simple, practical, and supported by existing clinical knowledge.

## INTRODUCTION

Globally, chronic respiratory diseases are among the top reasons for morbidity and mortality (*Alwafi et al., 2023*). Bronchial asthma (BA) is a common chronic respiratory disease that affects all age groups (*Rojo-Tolosa et al., 2023*). It is a heterogenous respiratory disease characterised by hyper-reactive and reversible airway inflammation, and its prevalence has recently increased worldwide. BA already affects 5–10% of the population (*Louis et al., 2022*). Severe BA affects approximately 5–10% of the asthmatic population (*Pefani et al., 2023*). Despite advances in medical technology, BA mortality has not improved significantly. BA is responsible for approximately one in every 250 deaths (*Althoff et al., 2020*). Acute respiratory failure (ARF) is a common complication in BA patients (*Chen et al., 2023*). BA patients with ARF have high rates of mechanical ventilation, extended hospital stays, and increased mortality (*Yii et al., 2019*). Therefore, early identification of patients with a high risk of ARF and a targeted management plan may significantly improve outcomes.

According to research, the high incidence of severe BA exacerbations and poor BA control has been linked to poor adherence to BA medications, poor BA symptom control, a low income, and a low education level (*Yan et al., 2016*). The risk factors for ARF in BA patients have not been the subject of a comprehensive study.

The equential organ failure assessment score (SOFA Score) is a commonly used disease severity scoring system which was developed to assess organ function in hospitalized patients. The Modified Early Warning Score (MEWS) is a five scaled assessment tool which was developed for the purpose of early identification of patients in need of a higher level of care and those at risk of complications and death (*Triantafyllidou et al., 2023*). According to a few studies, the SOFA Score and MEWS can identify patients with a high mortality risk in BA patients as an early warning score (*Klinger et al., 2021*; *Mitsunaga et al., 2019*; *Pölkki et al., 2022*). These scores are not specific to the BA patients and only assess the risk of death. Although ARF is related to death in BA patients (*Stather & Stewart, 2005*), these scores cannot be used to predict the occurrence of ARF. Currently, there is no clinically mature scoring system to predict the risk of ARF in BA patients.

The studies on BA complicated with ARF are imperfect. This study aims to develop a model to predict the probability of ARF in BA patients. The study seeks to construct the model in three ways to ensure performance and then selects the model with the best performance.

## METHODS

### Study design and population

The modeling group comprised BA patients hospitalized at the Second Affiliated Hospital and Yuying Children's Hospital of Wenzhou Medical University between 2017 and 2022. The validation group consisted of BA patients hospitalized at the First Affiliated Hospital of Wenzhou Medical University between 2019 and 2022. These are China's two large tertiary hospital. Our target population consisted of patients over 18 years of age who were admitted for BA but did not have ARF at admission. According to the diagnostic criteria

established by The Global Initiative for Asthma (GINA), the following inclusion criteria were used for the study: (1) All cases meetevery case met the diagnostic criteria for asthma; (2) clinical symptoms and characteristics of asthma were present; (3) medical history: recurrent wheezing, dyspnea, chest tightness or cough, primarily related to exposure to allergens, viral upper respiratory tract infection, physical and chemical stimulation, cold air, and exercise; (4) signs: During an attack, scattered or diffuse wheezing is audible in both lungs, primarily during the expiratory phase, and the expiratory phase was prolonged; (5) reversible: The above symptoms can be relieved by treatment or themselves. Exclusion criteria were as follows: (1) severe heart, liver, kidney, and other organ diseases; (2) upper airway obstruction (lung cancer, BTB, relapsing polychondritis, and tracheal foreign body); (3) allergic bronchopulmonary aspergillosis. The primary outcome was ARF during hospitalization. ARF was defined as breathing air with $PaO_2$ (Pressure of Oxygen) <60 mm Hg (one mm Hg = 0.133 kPa) and/or $PaCO_2$ (partial pressure of carbon dioxide in artery) >50 mm Hg. Only the first admission data was included when patients were admitted multiple times. Informed consent was exempted due to the retrospective format of this study. This study was approved by the Ethics Committee of the Second Affiliated Hospital and Yuying Children's Hospital of Wenzhou Medical University (Ethical Application Ref: 2021-K-148-01).

## Data collection

The following data were collected retrospectively for the study: (1) demographic information including age, male sex, body mass index (BMI), economic condition (poor: annual household income: RMB0-80, 000; middle: annual household income: RMB80,000-300,000; rich: annual household income: >RMB300, 000), education level (low level: elementary school and junior high school; middle level: high school, technical or vocational school; high level: university or college), smoking, smoking time, and smoking quitting time; (2) comorbidities including bronchiectasis, chronic obstructive pulmonary disease (COPD), emphysema, destroyed lung, thoracic deformity, cardiac insufficiency, valvular heart disease, cardiac enlargement, and area of pulmonary infection; (3) laboratory tests include albumin, prealbumin, C-reactive protein (CRP), white blood cell (WBC), and procalcitonin (PCT), (4) vital signs include temperature, heart rate, breath, mean arterial pressure (MAP), and oxygen saturation ($SpO_2$); (5) treatment includes onset-to-treatment time, hormone initiation time, the route of hormone administration, and long-term regular medication treatment (Table 1). The data on laboratory results were collected during the first examination before ARF (within 24 h of admission).

## Model development

Three models were developed in different ways: a statistics-driven model (model A), a clinical knowledge-driven model (model B), and a decision tree model (model C) to ensure the model's stability and validity. The three models' performance was compared.

Model A: the statistics-driven model. The model utilized statistical analysis to determine the model's predictive variables. The least absolute shrinkage and selection operator (LASSO) method and multivariate logistic regression analysis were applied to select

**Table 1 Candidate variables.**

| Domains | Variables* |
|---|---|
| **Demographics** | (1) Age, (2) Male, (3) Body mass index (BMI), (4) Economic condition, (5) Education level, (6) Smoking, (7) Smoking time, (8) Smoking quitting time |
| **Underlying diseases** | (9) Bronchiectasia, (10) Chronic obstructive pulmonary disease (COPD), (11) Emphysema, (12) Destroyed lung, (13) Thoracic deformity, (14) Cardiac insufficiency, (15) Valvular heart disease, (16) Cardiac enlargement, (17) Area of pulmonary infection |
| **Laboratory tests** | (18) Albumin, (19) Prealbumin, (20) C-reactive protein (CRP), (21) White blood cell (WBC), (22) Procalcitonin (PCT) |
| **Vital signs** | (23) Temperature, (24) Heart rate, (25) Breath, (26) Mean arterial pressure (MAP), (27) Oxygen saturation (SpO2) |
| **Treatment** | (28) Onset-to-treatment time, (29) Hormone initiation time, (30) The route of hormone administration, (31) Long-term regular medication treatment |

variables. LASSO regression is a variable selection method with high model stability. By using the penalty factor ($\lambda$) in the model, the regression coefficients are continuously compressed, and parts of them are compressed to zero to exclude these variables from the model. LASSO can effectively deal with multicollinearity and overfitting (*Jatuworapruk et al., 2020*). In the regression coefficient plot, each curve represents the change track of each independent variable coefficient (Fig. S1). LASSO regression was utilized to screen 12 variables. These variables were included in the multivariable logistic regression model. The variables with $P < 0.05$ were eliminated. Continuous variables were transformed into categorical variables based on quartiles and clinical reality. Then, multivariate logistic regression was performed on all categorical variables to determine the final predictor variables in the predictive scoring system. To correct for over-optimism, we further shrunk the regression coefficient by multiplication with a linear shrinkage factor. The shrinkage factor was calculated using the following formula: sf = [model $\chi^2 -$ (df $- 1$)]/model $\chi^2$. Model $\chi^2$ represented the $\chi^2$ value of the model, whereas df represented a degree of freedom (*Van Houwelingen & Le Cessie, 1990*). The prediction model was constructed based on the adjusted $\beta$ regression coefficient. $\beta$ regression coefficient represents the amount of change in the tested variable when the covariate is increased by one unit (*Razzaghi et al., 2013*). This model was developed by allocating an integer or half an integer score, assigning a value of 1 to the smallest regression coefficient, calculated by dividing the regression coefficient of each predictor variable by the smallest regression coefficient.

Model B: clinical knowledge-driven model. All variables were included in univariate logistic regression. The variables were preselected based on clinical knowledge and included in multivariate logistic regression. Seven significant variables obtained from multivariate logistic regression analysis were included in the model. Then, these continuous variables

were transformed into categorical variables, and multivariate logistic regression was performed to build a model based on the $\beta$ regression coefficients of the variables.

Model C: decision tree model. The decision tree algorithm is a machine learning algorithm. The C5.0 algorithm was used to construct the decision tree classifier, and the classifier was used to generate the intelligent tree model and the rules of the tree structure. The C5.0 decision tree algorithm allowed no assumptions regarding the statistical distribution of data, it can utilize non-parametric input data, and its application in the ARF prediction model was considered a reasonable choice for this study. The decision tree C5.0 algorithm used the information gain rate as the optimal segmentation criterion for candidate variables. The next variable was split according to the C5.0 algorithm to create a tree structure. The C5.0 algorithm used the post-pruning method to prune upward from leaf nodes, and the key issues were the error estimation and the setting of pruning criteria. Pruning was determined using the "error reduction" method. Initially, the weighted error of leaf nodes in the subtree to be pruned was calculated. Then, the error was compared to the error of the parent node. If the former error was greater than the latter, it could be subtracted. The result was represented by a decision tree, with each node depicting a different class of variables.

## Model performance comparison and model validation

The model performance was evaluated using discrimination, calibration, and precision. Discrimination refers to the ability to distinguish whether an outcome occurred or not. The area under the receiver operating characteristic curve (AUC) was used to examine the model's discrimination. AUC $\geq 0.8$ indicated excellent discrimination, AUC $\geq 0.7$ indicated acceptable discrimination, and AUC $< 0.7$ indicated poor discrimination (*Weng et al., 2021*). The AUC is summarized with the concordance statistic (C-statistic). The Hosmer-Lemeshow (H-L) goodness-of-fit test was used to verify the calibration degree of the model. The Hosmer–Lemeshow goodness-of-fit test showed $p > 0.05$, indicating no significant difference between predicted and observed probabilities. The overall performance of the prediction model was quantified as the Brier score, representing the mean squared difference between actual and predicted ARF, including discrimination and calibration. Brier scores ranged between 0 and 1, with lower scores indicating better model performance.

## Model validation

The bootstrap resampling method was used to validate, with 1,000 bootstrap samples internally. A bootstrap resampling sample with the same sample size is constructed by putting back sampling in the model development queue. The process is repeated 1000 times by using this sample as a training set and the model development queue as a verification set to evaluate the performance of the model. An ideal prediction model would have similar performance in the development and validation datasets (*Jatuworapruk et al., 2020*). Another dataset was used for the external validation of the predictive performance. Statistical analysis was performed using IBM SPSS Statistics software (V.26; SPSS Inc., Chicago, IL, USA) and R (version 3.6.1; *R Core Team, 2019*).

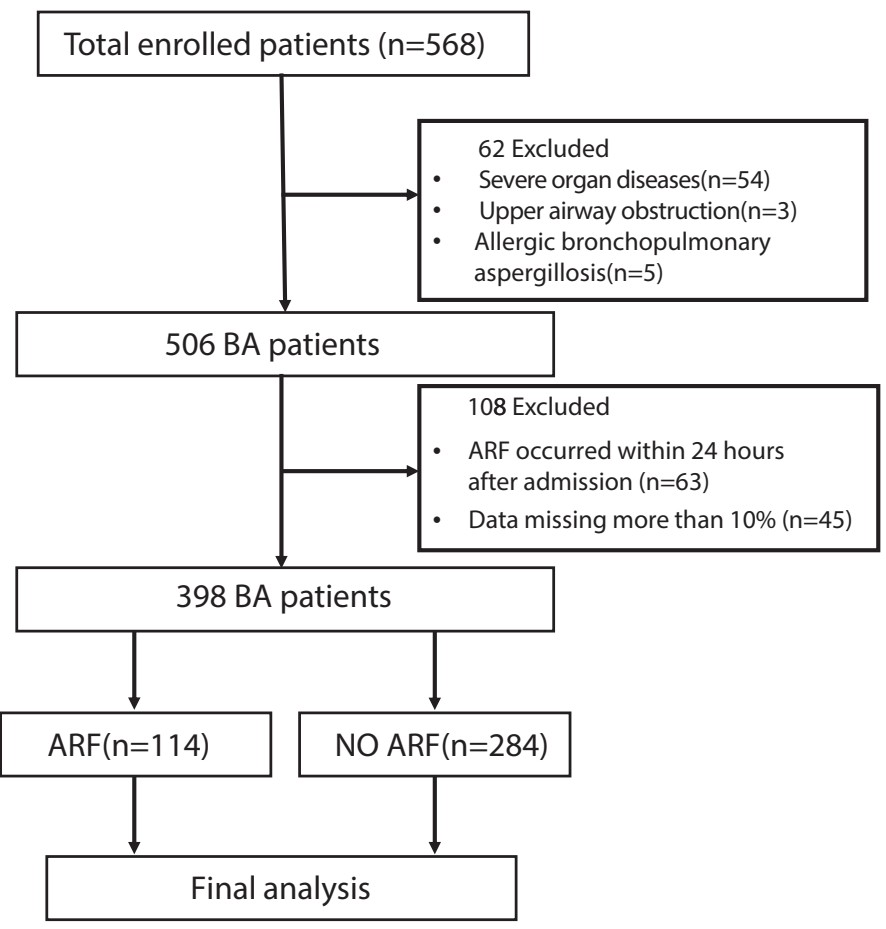

**Figure 1** **The study flow diagram.**

## RESULTS

### Cohort

The study flow diagram was shown in Fig. 1. This study included 398 patients in the final analysis, including 298 patients in the modeling group, 88 patients developed ARF, and 100 patients in the validation group, 26 patients developed ARF. The modeling group had a 30% ARF incidence, while the validation group had a 26% ARF incidence. Table 2 presents the characteristics of the cohort.

### Prediction models

We used three methods to generate three predictive models with different variables to predict ARF occurrence in BA patients during hospitalization. Three variables were consistently selected across all three models: economic condition, BMI, and albumin (Table 3). In model A, LASSO regression preselected twelve candidate variables. Figure S1 shows the results of LASSO regression. Finally, seven variables significantly associated with ARF were selected by multivariate logistic regression. Twelve predictive variables were

**Table 2  Demographic and clinical characteristics of the patients with asthma.**

| Variables | Total (n = 398) | Training set (n = 298) | Validation set (n = 100) | P- value |
|---|---|---|---|---|
| **Demographics** | | | | |
| Age, Median (years) | 66 (60, 72) | 66 (54, 73) | 66 (63, 68.25) | 0.78 |
| Male sex, n (%) | 198 (50) | 151 (51) | 47 (47) | 0.603 |
| BMI, Median (kg/m²) | 22.73 ± 4.16 | 22.71 ± 4.22 | 22.8 ± 3.98 | 0.855 |
| Economic condition, n (%) | | | | 0.617 |
| Poor | 30(8) | 23(8) | 7(7) | |
| Middle | 322(81) | 238(80) | 84(84) | |
| Rich | 46(12) | 37(12) | 9(9) | |
| Education level | | | | 0.311 |
| Low level | 56 (14) | 41(14) | 15(15) | |
| Middle level | 270(68) | 198(66) | 72(72) | |
| High Level | 72(18) | 59(20) | 13(13) | |
| Smoking, n (%) | 157 (39) | 121 (41) | 36 (36) | 0.486 |
| Smoking time, Median (years) | 0 (0, 10) | 0 (0, 20) | 0 (0, 8) | 0.408 |
| Smoking quitting time, Median (years) | 0(0,0) | 0(0,0) | 0(0,0) | 0.006 |
| **Comorbidities** | | | | |
| Bronchiectasia, n (%) | 96(24) | 75(25) | 21(21) | 0.479 |
| COPD, n (%) | 62(16) | 46(15) | 16(16) | 1 |
| Emphysema, n (%) | 86(22) | 65(22) | 21(21) | 0.976 |
| Destroyed lung, n (%) | 10(3) | 7(2) | 3(3) | 0.717 |
| Thoracic deformity, n (%) | 17(4) | 10(3) | 7(7) | 0.15 |
| Cardiac insufficiency, n (%) | 53(13) | 41(14) | 12(12) | 0.781 |
| Valvular heart disease, n (%) | 23(6) | 17(6) | 6(6) | 1 |
| Cardiac enlargement, n (%) | 54(14) | 44(15) | 10(10) | 0.301 |
| Area of pulmonary infection, Median (%) | 10 (3, 20) | 9 (3, 20) | 11 (3, 21.25) | 0.309 |
| **Laboratory tests** | | | | |
| Albumin, Median (g/dL) | 39.5 (36, 42.6) | 39.6 (35.9, 42.85) | 39.2 (36.45, 42.5) | 0.765 |
| Prealbumin, Median (mg/L) | 220 (161.25, 259) | 220 (161.25, 258.75) | 229 (170.25, 263.5) | 0.476 |
| CRP, Median (mg/dL) | 6.89 (0.97, 20) | 6.89 (0.73, 20) | 6.85 (1.96, 19) | 0.46 |
| WBC, Median (*109/L) | 8.01 (6.5, 11) | 8 (6.59, 10.44) | 0.451 | 0.451 |
| PCT, Median (ng/mL) | 0.05 (0.02, 0.12) | 0.05 (0.01, 0.12) | 0.05 (0.02, 0.13) | 0.87 |
| Vital signs | | | | |
| Temperature, Median (°C) | 37.1 (36.7, 37.3) | 37 (36.7, 37.2) | 37.2 (36.8, 37.6) | 0.004 |
| Heart rate; Median (bmp) | 89 (80, 100) | 88.5 (80, 100) | 90 (77.75, 101) | 0.779 |
| Breath, Median (bmp) | 20 (19, 22) | 20 (19, 21) | 21 (19, 22) | 0.643 |
| MAP, Median (mmHg) | 97 (89, 106) | 97 (89, 106) | 97.5 (91, 106) | 0.501 |
| SpO2, Median (%) | 96 (94, 97) | 96 (95, 97) | 95 (94, 96) | 0.003 |

**Table 2** (*continued*)

| Variables | Total (*n* = 398) | Training set (*n* = 298) | Validation set (*n* = 100) | *P*- value |
|---|---|---|---|---|
| **Treatment** | | | | |
| Onset-to-treatment time, Median(hours) | 70 (24, 115.25) | 70 (24, 120) | 70 (22, 101) | 0.854 |
| Hormone initiation time, Median(hours) | 68 (22, 95) | 50 (24, 96) | 68 (22, 93.25) | 0.511 |
| The route of hormone administration, n (%) | | | | 0.092 |
| No use | 1 (0) | 0 (0) | 1 (1) | |
| Intravenous drug use | 165(41) | 116(39) | 49(49) | |
| No intravenous drug use | 134(34) | 106(36) | 28(28) | |
| Both | 98(25) | 76(26) | 22(22) | |
| Long-term regular medication treatment, n (%) | 187(47) | 142(48) | 45(45) | 0.731 |
| Respiratory failure, n (%) | 114 (29) | 88 (30) | 26 (26) | 0.584 |

**Table 3  Three approaches for model development and their performance.**

| Approaches | Variables | Variables | Discrimination C- statistics (95% CI) | Internal validation C-statistics (95% CI) | Calibration H-L /P-value | Precision Birer score |
|---|---|---|---|---|---|---|
| Model A: the statistics- driven model. | 7 | Economic condition; BMI; Albumin; Area of pulmonary infection; MAP; Smoking; Long-term regular medication treatment | 0.856 (0.810–0.902) | NA | 10.365/0.240 | 0.1331 |
| Model B: clinical knowledge- driven model | 7 | BMI; Economic condition; Smoking; Albumin; Area of pulmonary infection; Hormone initiation time; Long-term regular medication treatment | 0.862 (0.820–0.904) | 0.854 (0.820–0.900) | 10.407/0.238 | 0.1320 |
| Model C: decision tree model. | 11 | Economic condition; PCT; Hormone initiation time, Albumin; BMI; Heart rate; Cardiac enlargement; Education level; WBC; Valvular heart disease; CRP | 0.846 (0.821–0.872) | NA | NA | 0.1350 |

retained based on clinical knowledge in the clinical knowledge-driven model (model B). Multivariate regression analysis confirmed the association of seven variables with ARF. Table 4 display the regression coefficients of each variable in model B. Table S1 provides the regression coefficients of each variable in model A. The decision tree produced eleven variables, and Fig. S2 described the relationship between the eleven predictive variables and ARF.

Models A and B have C-statistics of 0.856 (0.810−0.902) and 0.862 (0.820−0.904), respectively (Fig. 2, Table 3). Model B has a higher discrimination degree and is convenient. The C-statistics of model C was 0.846 (0.821−0.872), and the model contained eleven variables, and the practicability was poor. The Hosmer-Lemeshow test revealed that model A and model B had good calibration (Table 3). The Birer scores of models A, B, and C were 0.1331, 0.1320, and 0.1350, respectively (Table 3). According to the Hosmer-Lemeshow test, model A and model B had good calibration in external verification, and the three models' Brier scores were 0.0645, 0.0640, and 0.0773, respectively (Table S2). Brier scores were similar for the three models. In external verification, model A and B also had similar C-statistics of 0.897 (0.822−0.971) and 0.890 (0.815−0.965), respectively (Table S2, Fig. 2). According to the clinical practice, the model's performance, and the clinical significance of each variable, we finally choose the clinical knowledge-driven model (model B) (Fig. 3, Table 3).

**Table 4** Variables in the final model using multivariable regression with shrinkage (clinical knowledge- driven model; model B).

| Variable | Regression coefficients | Regression coefficients with shrinkage | OR (95% CI) | P-value | Score |
|---|---|---|---|---|---|
| BMI | | | | <0.001 | |
| <25 | Reference | | | | |
| ≥25~<30 | 1.494 | 0.489 | 4.453 (2.120–9.354) | <0.001 | 2 |
| ≥30 | 1.853 | 0.607 | 8.382 (1.435–28.387) | 0.015 | 2.5 |
| Economic condition | | | | | |
| Rich | Reference | | | | |
| Not rich (poor and middle) | 1.829 | 0.599 | 6.227 (1.256–30.886) | 0.025 | 2.5 |
| Smoking | 1.187 | 0.389 | 3.278 (1.718–6.256) | <0.001 | 1.5 |
| Albumin | | | | | |
| >40 | Reference | | | | |
| ≤40 | 1.194 | 0.391 | 3.301 (1.649–6.608) | 0.001 | 1.5 |
| Area of pulmonary infection | | | | 0.004 | |
| 0 | Reference | | | | |
| >0~≤20 | 1.254 | 0.411 | 3.503 (1.020–12.037) | 0.046 | 1.5 |
| >20 | 2.056 | 0.673 | 7.812 (2.122–28.765) | 0.002 | 2.5 |
| Hormone initiation time | | | | | |
| ≤72 | Reference | | | | |
| >72 | 0.795 | 0.260 | 2.214 (1.146–4.277) | 0.018 | 1 |
| No long-term regular medication treatment | 1.442 | 0.472 | 4.229 (2.157–8.291) | <0.001 | 2 |
| Total | | | | | 13.5 |

**Notes.**
Shrinkage factor:0.3274.

Finally, we performed bootstrap validation with 1000 samples for the selected model B. The optimism corrected C-statistics was 0.854 (0.820−0.900) (Table 3). The clinical knowledge model retained good discrimination and calibration in the 1,000-time bootstrapping test set.

### The A$_2$-BEST$_2$ risk score

According to a clinical knowledge-driven model (model B), seven predictive variables were used to develop the risk score for ARF occurrence in BA patients during hospitalization: the A$_2$-BEST$_2$ risk score (A$_2$ (area of pulmonary infection, albumin), BMI, Economic status, Smoking, and T$_2$ (hormone initiation time and long-term regular medication treatment)). Each variable was scored according to its regression coefficient (Table 4). The risk score for ARF in each BA patient was calculated according to the score corresponding to each variable. The A$_2$-BEST$_2$ risk score ranged from 0 to 13.5. BA patients were divided into four groups based on their scores: low-risk (0–4 points), moderate-risk (4.5–8 points), high-risk (8.5–11 points), and very high-risk (11.5–13.5) groups (Table 5, Fig. 3). In the four groups, the predicted probabilities of ARF were 1.37%, 14.92%, 57.40% and 90.32%, respectively. In different risk groups, the actual probability of occurrence was close to the predicted probability. The risk stratification of the score placed 44 (14.8%) cases in

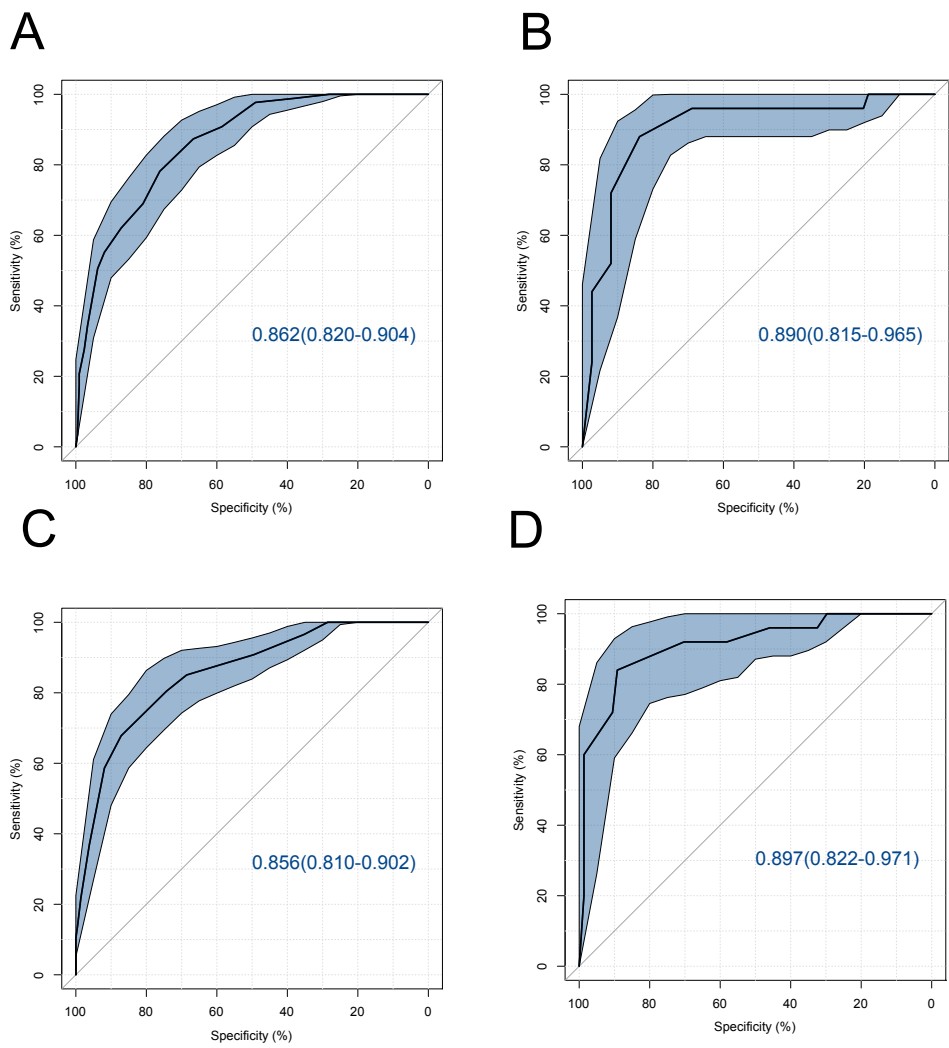

**Figure 2  Receiver operating characteristic curves of the clinical knowledge-driven model in predicting ARF in modeling group (A) and external validation group (B). Receiver operating characteristic curves of the statistics-driven model in predicting ARF in modeling group (C) and external validation group (D).**

the low-risk group, 153 (51.3%) cases in the moderate-risk group, 81 (27.2%) cases in the high-risk group, and 20 (6.7%) cases in the high-risk group. Furthermore, the ARF probability predicted by risk score was very close to that of actual ARF probability in the different risk groups (Table 5). We compared the predictive power of the prediction model with Sofa score and MEWS. The area under the ROC curve of Sofa score and MEWS for predicting ARF in BA patients was 0.668 (0.598−0.737) and 0.628 (0.555−0.701), respectively (Fig. S3). The A2-BEST2 Risk Score predictive capability was superior to the Sofa score and MEWS.

| BMI ( kg/㎡ ) | |
|---|---|
| <25 | 0 |
| ≥25~<30 | 2 |
| ≥30 | 2.5 |

| Economic condition | |
|---|---|
| rich | 0 |
| middle | 2.5 |
| poor | 2.5 |

| Albumin(g/dL) | |
|---|---|
| ≤40 | 1.5 |
| >40 | 0 |

| Area of pulmonary infection(%) | |
|---|---|
| 0 | 0 |
| >0~≤20 | 1.5 |
| >20 | 2.5 |

| Smoking | |
|---|---|
| YES | 1.5 |
| NO | 0 |

| Hormone initiation time (h) | |
|---|---|
| >72 | 1 |
| ≤72 | 0 |

| Long-term regular medication management | |
|---|---|
| YES | 0 |
| NO | 2 |

| Risk score | Risk group | ARF risk |
|---|---|---|
| 0-4points | Low | 1.37% |
| 4.5-8 points | Moderate | 14.92% |
| 8.5-11 points | High | 57.50% |
| 11.5-13.5 points | Very high | 90.32% |

**Figure 3** The $A_2$-BEST2 risk score and the risk groups low risk 0–4 points, moderate risk 4. Points: 5–8 points, high risk 8.5–11 points, very high risk 11.5–13.5 points.

**Table 5 Risk of ARF in clinical knowledge-driven model according to risk stratification.**

| Risk stratification | N (%) | Predicted incidence (%) | Actual incidence (%) |
|---|---|---|---|
| Low (0–4) | 44 (14.8) | 1.37 (1.15–1.58) | 0 |
| Moderate (4.5–8) | 153 (51.3) | 14.92 (13.50–16.35) | 18 |
| High (8.5–11) | 81 (27.2) | 57.50 (54.10–60.72) | 53 |
| Very high (11.5–13.5) | 20 (6.7) | 90.32 (88.28–92.38) | 90 |

## DISCUSSION

This study used two retrospective cohorts admitted at a comprehensive tertiary hospital as the study subjects. We constructed a novel and never studied predictive model for predicting ARF during hospitalization in BA patients: the $A_2$-BEST2 risk score. The predictive model integrates laboratory and demographic data and treatment variables to accurately predict the risk of ARF in BA patients. The clinical knowledge-driven model has better performance and practicability than the decision tree and data-driven models. The score can be easily implemented based on the available common variables and has good performance in the development and validation cohorts.

We selected seven predictive variables from the clinical knowledge-driven model (model B) related to ARF occurrence in BA patients. In this study, seven variables, such as area of pulmonary infection, albumin, BMI, economic condition, smoking, hormone initiation time and long-term regular medication treatment, were associated with ARF in BA patients (Table 3, Fig. 3). Current studies have not discovered the risk factors and prediction of ARF in BA patients. This study may partially address this research gap and contribute to this topic's literature.

Obese patients had poorer expiratory reserve volume and functional residual capacity than normal-weight BA patients (*Brazzale, Pretto & Schachter, 2015*). Obese patients have lower forced expiratory volume in one second (FEV1) and forced vital capacity (FVC), and environmental factors greatly affect their respiratory symptoms (*Kasteleyn et al., 2017*). Several studies have demonstrated that obese patients with BA have less airway inflammation than patients with normal BMI, have lower fractional exhaled nitric oxide (FeNO), with lower the fractional exhaled nitric oxide (FeNO) and have a poor response to corticosteroid therapy (*Berg et al., 2011*; *Sivapalan, Diamant & Ulrik, 2015*). There are restrictive functional changes in obese patients with asthma (*Kasteleyn et al., 2017*). In addition to these factors, BA patients with high BMI indexes are more likely to develop ARF. This suggests that early weight management in BA patients is important to prevent ARF progression.

Income and commercial insurance may affect patients' compliance with specific treatment methods. The study revealed that poor people might have limited access to treatments and drugs, resulting in inadequate control of BA (*Maddux et al., 2021*). Epidemiological investigations indicate that BA incidence is also high in poor families and developing countries (*Sinclair et al., 2018*). The high incidence of severe BA attacks is due to poor compliance with drugs and non-regular treatment (*Yan et al., 2016*). Therefore, economical and reasonable treatment programs are important to improve treatment compliance among poor people. It can reduce ARF incidence in BA patients during acute attacks of BA and reduce hospitalization costs to a certain extent.

Smoking is related to the severity of asthma. Smoking and exposure to smoke can increase airway inflammation and responsiveness, making severe asthma easier to develop in patients (*Singh & Busse, 2006*). Air pollution, such as carbon monoxide and organic carbon, increases the risk of exacerbation of asthma. Carbon monoxide is a primary component of cigarette smoke (*Norris et al., 2000*). Early smoking cessation is critical in controlling BA and preventing ARF.

Few studies have examined the relationship between serum albumin levels and ARF in BA patients. The risk of ARF and mechanical ventilation increases as admission albumin levels fall below the normal range in general patients (*Thongprayoon et al., 2020*). In our study, patients with low albumin BA had a higher risk of ARF. Albumin plays a role in maintaining lung function and asthma stability when air quality is poor. The non-enzymatic antioxidant and anti-inflammatory properties of unified albumin play a role in maintaining lung function (*Khatri et al., 2014*). This may be why BA patients with low serum albumin are more likely to develop ARF.

Corticosteroid therapy is a primary treatment for an acute attack of BA. Studies have shown that early use of corticosteroids in treating BA attacks can reduce the admission rate of patients. This treatment appeared to benefit most patients with severe illness and those who have not recently received oral hormone therapy (*Rowe et al., 2001*). This also proves that early corticosteroid use can reduce the risk of exacerbation of BA and ARF to some extent, consistent with our findings.

Currently, research on the risk factors and prediction models of ARF in BA patients is insufficient. The research focuses primarily on the risk factors and prediction models of

BA attacks (*Lowden & Turner, 2022*). To our knowledge, no risk score systems have been developed based on integers or fractional integers. The prediction model was constructed in three different ways in this retrospective study. A contraction technique was applied in models A and B to correct excessive optimism. We used a cohort from a different hospital at the same level as a validation cohort, along with internal validation, because the sample size and positive outcome events of the development cohort were relatively small. The model's internal and external validation performance was good, yielding similar results. This strengthens the evidence for our findings. The clinical knowledge-driven variable selection is superior to the statistics-driven model. A clinical knowledge-driven model can assist clinicians in understanding and utilizing predictive variables. Our study also has some limitations. When the BMI variable was changed from a continuous variable to a categorical variable, none of the patients with BMI<18.5 developed ARF, so all the patients with BMI <25 were classified as the reference group. This study's sample size was small. The possibility of residual confounding cannot be excluded, and all existing predictors cannot be captured because this is a retrospective study. This study's lengthy duration may increase the heterogeneity of the enrolled patients due to treatment and environmental changes. Therefore, larger prospective studies are needed to validate our findings further and to optimize the prediction model.

## CONCLUSION

Our study provides evidence to support the role of variables that are assumed to increase the risk of exacerbation of ARF in BA patients. This study constructed the prediction model in three different ways. Finally, we selected the clinical prediction model as the final use model, with better prediction ability and calibration. It addresses the lack of a prediction model in current clinical research.

## ACKNOWLEDGEMENTS

We want to thank all participants for their support and contribution to this study. Without their voluntary participation and their openness to discuss their personal experiences and opinions, it would not have been possible to achieve these results.

### Funding
This study was supported by the Wenzhou Municipal Science & Technology Bureau, China. (Grant No:Y20210840). The funders had no role in study design, data collection and analysis, decision to publish, or preparation of the manuscript.

### Grant Disclosures
The following grant information was disclosed by the authors:
The Wenzhou Municipal Science & Technology Bureau, China:  Y20210840.

## Competing Interests
The authors declare there are no competing interests.

## Author Contributions
- Yanhong Qi conceived and designed the experiments, prepared figures and/or tables, authored or reviewed drafts of the article, and approved the final draft.
- Jing Zhang analyzed the data, authored or reviewed drafts of the article, and approved the final draft.
- Jiaying Lin analyzed the data, authored or reviewed drafts of the article, and approved the final draft.
- Jingwen Yang performed the experiments, authored or reviewed drafts of the article, and approved the final draft.
- Jiangan Guan performed the experiments, authored or reviewed drafts of the article, and approved the final draft.
- Keying Li performed the experiments, authored or reviewed drafts of the article, and approved the final draft.
- Jie Weng analyzed the data, prepared figures and/or tables, and approved the final draft.
- Zhiyi Wang conceived and designed the experiments, prepared figures and/or tables, and approved the final draft.
- Chan Chen conceived and designed the experiments, prepared figures and/or tables, and approved the final draft.
- Hui Xu conceived and designed the experiments, prepared figures and/or tables, authored or reviewed drafts of the article, and approved the final draft.

## Ethics
The following information was supplied relating to ethical approvals (i.e., approving body and any reference numbers):

The Ethics Committee of the Second Affiliated Hospital and Yuying Children's Hospital of Wenzhou Medical University (Ethical Application Ref: 2021-K-148-01).

## Data Availability
The raw date are available in the Supplemental Files.

## Supplemental Information
Supplemental information for this article can be found online at http://dx.doi.org/10.7717/peerj.16211#supplemental-information.

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
