# Peer review of "Predicting the risk of acute respiratory failure among asthma patients—the A2-BEST2 risk score: a retrospective study"

_PeerJ, doi:10.7717/peerj.16211_

## Round 0.1 · original submission · Major Revisions

Reviewer 1 has provided valuable feedback and suggestions for improving the manuscript. They have pointed out areas in the introduction, experimental design, and validity of the findings sections where more information is needed. They have also requested clarification and additional explanations for various aspects, such as the significance of certain variables, statistical methods used, interpretation of figures and tables, and the rationale behind certain decisions made in the study. The reviewer also highlighted the need for improved clarity and accessibility for readers unfamiliar with the field and noted the issue with the visibility of initial words in Tables 3 and 4.

Reviewer 2 acknowledged the well-designed experiments and satisfactory writing but recommended minor revisions. They suggested updating the bibliography with references from 2023 and urged the authors to compare their model with existing risk scoring models in the literature. Additionally, they inquired about the impact of the duration of the disease on the described model, considering bronchial asthma is a chronic condition.

Based on the feedback from both reviewers, it is clear that revisions and additions are necessary to address the concerns raised. The manuscript should be revised to include more explanations and clarifications, especially regarding the introduction of key concepts, statistical methods, interpretation of results, and comparisons with existing models. It is also important to improve the accessibility and readability of the manuscript for readers outside the field. The issues with the visibility of text in Tables 3 and 4 should be rectified. Finally, the bibliography needs to be updated with recent references.

Overall, the manuscript has the potential to be accepted for publication with the recommended revisions.

Reviewer 1 ·

Basic reporting

Need to explain bit more about Bronchial asthma, ARF, SOFA score, MEWS in introduction/review part.

Experimental design

Insufficient information which need to improve
1. In line 92, explain about PaO2, PaCO2
2. In line 96, remove the
3. In line 118-119, need to explain about LASSO and multivariate logistic regression analysis and its application
4. In line 124, explain a bit about shrinkage regression coefficient
5. In line 128, what is β regression coefficient why it was included for study
6. In line 155, explain about Hosmer-Lemeshow goodness-of-fit-test
7. In line 161, explain about Bootstrap method

Validity of the findings

1. In line 166-169, explain more about figure 1. figure 1 shows 398 BA patients with ARF than again it was divided in two groups ARF (114) and NO ARF (284) its confusing 398 BA patients includes both ARF and NO ARF
2. In line 167-168, mentioned about 289 constituting modeling group and 100 constituting validation group. And its ARF incidence. How it was calculated and which figure or tables show these outcomes
3. In line 172-173, at the end of sentence, Only three variables……. Add figure or table number relevant to it
4. In line 173-174, mentioned about 12 variable candidates then 7 was selected from it so how it was done and also explain supplementary figure 1 what does it shows?
5. In line 177, Table 4 explain about regression coefficient and below table 4 (line 17) Shrinkage factor:0.3274 was mentioned, what does it show?
6. In line 180, provide interpretation of Supplementary figure 2
7. In line 181, explain briefly about C-statistics and its application
8. In line 182-183, mentioned that C model contains 12 variables but table 3 shows 11 variables why so?
9. In line 183-184, at the end of sentence of Hosmer……. Add relevant figure/table
10. In line 193, C-statistics was 0.854(0.820-0.900), in which table/figure it was shown?
11. In line 203-204, predicted probability of ARF for high Risk stratification75.4% was not shown table 5. Is this value correct?
12. Line 218-221, related to which figure/Table?
13. Table 4, total of score is 13.5, how did you get this value after sum of all score? If so its total is more than 13.5

Additional comments

Manuscript should be written in a way that who are not more into that field can readily understand it.
In Table 3 and 4, in first column text, initial words are not visible.

Reviewer 2 ·

Basic reporting

The article is nicely written. Use of English vocabulary is satisfactory. The authors has reached to certain result which they wanted to achieve.
However, the bibliography may be updated as no reference of 2023 is mentioned.

Experimental design

Well designed experiments. data was thoroughly investigated.

Validity of the findings

There are other similar risk scoring models are available in literature. The author must include comparison with those and justify how their model is better / unique.
Bronchial ashtma is a chronic disease. How the duration of the disease affects the model described here?

---

## Round 0.2 · accepted · Accept

Based on the feedback from both reviewers, it appears that your manuscript has been well-written and revised to address their suggestions. The experimental design and validity of the findings are also deemed acceptable.

However, there is a minor suggestion from Reviewer 1 regarding correcting the C-statistics value in line 215. The suggested value is 0.854 with a confidence interval of 0.820-0.904.

Given that both reviewers have provided positive feedback and there is only one minor correction to make, I would recommend accepting the manuscript for publication with the condition that the C-statistics value in line 215 is corrected as suggested by Reviewer 1. Once this correction is made, the manuscript should be ready for publication.

Reviewer 1 ·

Basic reporting

The article is well written, to understand the findings of the manuscript. All reviewer's suggestions have been incorporated into the reviewed manuscript.

Experimental design

Everything is well explained.
All reviewer's suggestions have been incorporated into the reviewed manuscript.

Validity of the findings

A simple predictive model is constructed for ARF and BA patients.

Additional comments

a minor suggestion

in line 215: correct C-statistics value. it should be 0.854(0.820-0.904)

Reviewer 2 ·

Basic reporting

revised ms is self explanatory and is clear to understand.

Experimental design

acceptable.

Validity of the findings

acceptable

Additional comments

nil